# Design of an Auxiliary Artificial Lymphatic Vessel in Treatment of Secondary Lymphedema Due to Breast Cancer

**DOI:** 10.3390/healthcare10010068

**Published:** 2021-12-31

**Authors:** Gabriela Durán-Aguilar, Alberto Rossa-Sierra, Rita Q. Fuentes-Aguilar

**Affiliations:** 1Facultad de Ingeniería, Universidad Panamericana, Álvaro del Portillo 49, Zapopan 45010, Mexico; gaduran@up.edu.mx; 2Tecnológico de Monterrey, Escuela de Ingeniería y Ciencias, Zapopan 45201, Mexico; rita.fuentes@tec.mx

**Keywords:** lymphedema, lymphatic vessel, medical devices

## Abstract

Breast cancer is the most common malignant tumor that affects women in the United States, Europe, and Mexico. As an adverse effect when performing treatments for this condition, secondary lymphedema associated with breast cancer occurs in some cases. This complication occurs due to the interruption of lymphatic flow in the upper extremities in conjunction with other factors such as radiation, sedentary lifestyle, removal of lymph nodes, damage to lymphatic vessels, and others. This article reviews breast cancer incidence, mortality, and survival patterns, confirming that, specifically, lymphedema has high health, social, and economic impacts. Research demonstrates that it fundamentally affects women at an early age. In approximately a third of the cases, it becomes a chronic disease. Therefore, physical therapy is essential for a better quality of life in patients who survive this disease. Surgeries and manual and pharmacological treatments are the current procedures done to reduce to reduce the alterations suffered by patients with lymphedema; however, the success of the treatments depends on each patient’s characteristics. To face this problem, the design of a lymphatic vessel has been proposed to assist the mechanical failure of the damaged lymphatic system. In this work, the design methodology used for the blueprint of the lymphatic vessel is presented, as well as the computer analysis of fluid simulation and the selection of the proposed material, resulting in the production of a micrometric design. In the future, it is expected that a surgeon will be able to implant the design of the vessel to restore lymph flow through the lymphatic system, thus helping to combat lymphedema.

## 1. Introduction

In 2020, according to National Institute of Statistics and Geography (INEGI) [1], the incidence of malignant breast tumors in the population over 20 years old was 14 new cases per 100,000 habitants. For women, it peaks in the 60–64 age group (69 cases per 100,000 women).

As a result of this analysis and the continuous contact with people who suffer from this condition, several aspects of breast cancer were observed both psychologically and physically. Depending on the type of cancer detected, the treatment that doctors prescribe is different; however, when there is surgery or radiation in the procedure, lymph nodes and lymphatic vessels are removed or damaged, in most cases causing the chronic disease known as lymphedema [2].

According to the World Health Organization, breast cancer is the most common cancer in women and is increasing in both developed and developing countries [3]. In 2020 alone, 2.3 million women were diagnosed with breast cancer, and 685,000 died from the disease [4]. Large numbers of cancer patients around the world lack access to quality diagnosis and treatment [5]. With breast cancer surgery and treatments, patients present physical symptoms, lymphedema, psychological affectations, family and professional role changes, among others [6], thus altering their quality of life. Multiple studies [5,6,7,8,9] indicate that once the most critical phase of breast cancer treatment has passed (surgeries, chemotherapies, radiotherapies, etc.), patients are forced to find a way to adapt to different situations that their new reality presents them [8].

American Cancer Society studies [10] observed that lymphedema could occur, among other reasons, during some of the surgical procedures for cancer. The surgeon may remove lymph nodes near the tumor to see if cancer has spread; when this happens, the lymphatic vessels that carry fluid from that area to the rest of the body are also extracted. Removal of lymph nodes and vessels makes it difficult for lymphatic fluid to flow from the arms, legs, and other body parts to the chest, where it is released back into the bloodstream [11]. If the remaining lymphatic vessels cannot take in enough fluid from the area, it accumulates, causing swelling of the affected part.

Another treatment that could cause lymphedema is radiation therapy, which can affect the passage of lymphatic fluid by causing damage and scar tissue in lymph nodes and vessels. Sometimes the tumor can block part of the lymphatic system and cause lymphedema and infections that limit the passage of fluids [12].

There is no surefire way to prevent all cases of lymphedema. Still, there are ways to lower the risk of developing the condition. For example, modification of risk factors, therapeutic exercise inclusion, compression therapy, skin care, weight control, and comorbidities. As risk preventive elements that are relevant [13,14,15], when it is necessary to remove lymph nodes, some modern surgical techniques can be useful due to the procedure performed by the doctor when selecting only the infected nodes; with this type of method, the extraction is safer and less aggressive for the patient [16].

According to the International Society of Lymphology [17], the severity of lymphedema is categorized into four types or clinical stages: Stage 0, subclinical or latent conditions. There is no obvious edema, but the lymphatic transportation is impaired; it may be months or years before edema become apparent. Stage 1, precocious accumulation of reversible proteinaceous fluid with limb elevation or compression therapy; may occur pitting edema. Stage II Early, elevation/compression alone no longer solves edema and pitting edema (e.g. Figure 1) Stage II Belated, edema with or without pit; there is fibrosis development in the tissues. Stage III, lymph static elephantiasis; the tissue is firm (fibrotic), there is absence of pit, acanthosis, fat deposits, warts, and other trophic skin changes.

When lymphedema occurs, the standard volume of lymph produced daily by the body is greater than the patient’s lymphatic system capacity; therefore, the body cannot prevent fluid accumulation, resulting in edema.

There are different treatments for the management or reduction of lymphedema, which include: mechanical treatments, manual lymphatic drainage, bandages, physiotherapy, among others [18,19]. Surgical treatments include early-stage lymph venous bypass (obstructed lymphatic vessels are connected by anastomosis to small nearby veins to improve lymphatic drainage; the efficacy is reflected around 12 months after surgery) [20] and lymph node transfer vascularized lymph nodes (healthy lymph nodes from an unaffected region are transplanted with a flap, it can offer a new type of permanent drainage, the efficacy is observed 6 to 9 months after surgery); in both cases the cure is infrequent, despite performing both procedures during the same surgery. In the case of severe lymphedema, liposuction is performed to reduce the amount of fat in the affected area. Due to the technique performed in surgeries, the medical staff has specialized training; it requires a particular piece that connects the lymphatic vessels, so it is not recommended to patients by surgeons [21].

### Problem Statement

Lymphedema can become evident more than 20 years after it initiated; it is a chronic disease that once diagnosed it can be treated but not cured, and therefore it becomes a lifelong condition for patients [22].

Not all patients are candidates for surgical treatment, some with advanced lymphedema may have severely damaged lymphatic vessels; in these cases, reduction surgery by liposuction becomes the effective and palliative treatment [23]. As long as the patient receives the diagnosis in a prompt and timely manner, manual treatments are the solution; however, if even with these treatments the lymphedema does not subside, the patient will be suitable for surgery. Once surgery is performed, there is a successful result in reducing lymphedema; however, in all cases, the use of compression garments will be imminent and once again palliative [23].

## 2. Method

Given the complications derived from the lack of treatment and cure for lymphedema, an implantable device has been designed to allow the maintenance of lymph flow after lymph node removal surgery. For this design development, the double diamond design model [24] was used, starting by analyzing a lymph node and its functions, as well as possible types of movement of objects related to the transportation of liquids in a unidirectional way.

For this type of project, the use of the tool known as Value Engineering [25] is proposed, which is employed for the functional understanding of products, services, and processes. The usage of functions is an essential support to analytical thinking, since it helps to understand what each of the parts that make up the object of study does, thus facilitating the transition to creative thinking for proposed solutions [25].

The function analysis study was carried out on two occasions: the first study was applied to the lymph node to identify the role that it performs within the body. As consequent to axillary dissection, the fulfillment of these functions will be affected totally in the extraction area and partially in the lymph circulatory system. Based on the requirements and specifications raised from the second study, the functions necessary for progress in the design process were extracted.

From this study, the following observations were obtained:Find a permanent function: connect parts. When the lymph node is removed, and by cauterization, the union between the lymphatic vessels is interrupted, preventing the flow of the lymph, which causes its accumulation.The essential functions, built based on the specific purpose of a lymphatic vessel, are twofold: allow the flow of the lymph and prevent the return of the fluid.

Determining the functions from the requirements (Table 1) will allow for the understanding of the purpose of the medical device. It will also open the possibility of designing a solution. For example, for manufacturing an intracorporeal device, the identification of functions and biological conditions must be considered. Alternative materials that can fulfill these functions may be sought to compare them, choosing the one that guarantees a better value to the product [26].

The Function Analysis System Technique (FAST) is used to analyze the functional structure of a technical system, displaying and separating the functions of a product to analyze them in isolation [26]. Figure 2 shows how the modification of objectives in the actions is necessary for the device. The essential functions registered will be connecting parts and dosing substances. The material must be a permanent function, complying with the specifications and regulations that ensure the medical phase.

### Project

As a result of the application of the Value Engineering Technique, the design problem was defined (Figure 2) considering the following:The stages of lymphedema selected are the most severe, with the lymphatic vessels damaged without possible reconstruction.Existing treatments such as anastomosis were analyzed, suggesting a connection to a thicker vein, with a different pressure (−6 to 0 mmHg) than that of a lymphatic vessel (25 and 32 mmHg). Therefore, with time, fibrosis is generated, again preventing the transit of lymph.In the case of lymph node transposition surgery, different kinds of interventions are required. One of these surgeries is used to remove lymphatic vessels in an unaffected limb and connect these vessels to the affected limb, causing a possible problem in the lymphatic flow.Obtain the lymphatic transit, connecting the lymphatic vessels through one or more artificial devices that allow the same pressure as the original 25 and 32 mmHg.The material required for the device must meet the compatibility specifications and not be an allergen for the body.The measures proposed for this model are the following: incoming valve >50–75 microns, outgoing valve <100 microns, able to resist a pressure of 25 and 32 mmHg, with an unidirectional flow of the lymphatic vessel of 2000 microns of length with a diameter of 20 microns.

Figure 3 shows the test of the proposed version 1.0. A curved design of the lymphatic vessel was not a requirement since it can also function correctly with a straight arrangement; it was made of a flexible material to allow muscle and whole-body movement. The pressure of the lymph and its viscosity are sufficient for its flow within it. An important consideration was to prevent the return of the lymph into the artificial lymphatic vessel.

In version 2.0, the measurements used for the geometry of the lymphatic vessel were as follows: possible opening of valves of 50–100 microns when the closure is generated, thickness of the walls is 20 microns, and the lymphatic vessel length is 800 microns.

Figure 4 shows how version 2.0, due to its geometric design, allows the flow of the lymph and prevents the return by exerting pressure on it. The problem found in the design is that there is no way to connect the artificial lymphatic vessel to any of its ends. The thickness of the wall will have the risk of closing if a suture is made for its connection. The results are in version 3.0. which can be seen in Figure 5. In this version, modifications were made to achieve the connection of the lymphatic vessel at its ends using a fundamental technique in vascular surgery, which will be explained in detail further on.

Figure 6 shows the left end (A) of the artificial lymphatic vessel, which has a smaller diameter than the right end (B) thereof, allowing the artificial lymphatic vessel in its modular version to connect the vessels between them. The slight low relief on the right end allows suturing and marks the surgeon’s limit of the lymphatic vessel and its internal valve (C).

For the design of the connection in the prototype of the lymphatic vessel, one consideration was vascular surgery using a basic technique. This surgery is known as vascular dissection, which allows the tissues to be taken and united with slight displacement [23]. Based on the surgery, the suture area was considered in the design.

## 3. Material Selection

The analysis of three types of polymers with physical properties and biological compatibility to avoid rejection, in approximation of the vein, was carried out for material selection. The material that provides a better approach to the general, physical, and mechanical properties of the vein/artery, analyzed for the replacement of the lymphatic vessel, is thermoplastic polyurethane with a hardness degree of Shore A. In the analysis carried out with the materials and the required needs of the lymphatic vessel, it was defined that Thermoplastic elastomer (TPE Mediprene 500M Shore 0A from Hexpol TPE^®^) complies with the mechanical properties for the correct functioning of the lymphatic vessel design, as well as with its biocompatibility and medical-grade that guarantees the standards to which it has been submitted.

### Nano Printing

The proposed process for the manufacture of the lymphatic vessel design is nano printing. The smallest size of the design is one micron, which is the most suitable for its production. Nano printing consists of the non-linear absorption of two photons to create a solid structure printed in 3D from a photoactivatable material [27].

## 4. Results

### 4.1. Fluid Simulation Analysis

#### 4.1.1. First Analysis with Version 1.0

The artificial lymphatic vessel design validation was developed using FEA (finite element analysis). This stage was carried out with SOLIDWORKS^®^ software, using the Flow Simulation package. For the materials used in calculations, considerations about the mechanical properties (thermoplastic elastomer with a hardness degree Shore A) were taken; the lymph viscosity is similar to blood and glycerin, with the lymph pressure of 8mm Hg at entry over the lymphatic vessel and with an outlet of 14 mmHg.

As shown in Figure 7, the pressure exerted on the lymphatic vessel allows the valve to open, granting lymph circulation. In Figure 8 it can be observed that the pressure exerted on the lymphatic vessel is slight; it prevents the passage of lymph and its correct return.

#### 4.1.2. Second Analysis with Version 2.0

In the second analysis, the same physical properties of the material and the blood were considered to simulate the lymph. The change was in the pressure of the lymphatic vessel inlet and outlet, as the hydrostatic pressure required capillary ranging from 25 mmHg to 32 mmHg. In the first analysis, the indirect capillary hydrostatic pressure was mistakenly considered (Figure 9 and Figure 10).

### 4.2. Verification

Third analysis with version 3.0.

With hydrostatic capillary pressure from 25 mmHg to 32 mmHg, an increase in velocity is observed after flowing through the valve compared to the velocity at the inlet. This increase in speed is due to the decrease in the diameter through which the lymph flows (Figure 11 and Figure 12). Regarding the volume flow, it can be observed that it remains constant on average except for minor fluctuations. This constant flow is to be expected because it is a closed container; and following the law of conservation of mass, the flow in the outlet must be the same as in the input (Figure 13 and Figure 14).

In the area with the largest diameter, the pressure value is greater than the value at the outlet where the diameter is smaller. This pressure difference makes it possible for there to be a flow of lymph, where it goes from an area of higher pressure to one of lower pressure. (Figure 15 and Figure 16). Final lymphatic vessel design (Figure 17).

## 5. Conclusions

The physical and mechanical properties required for the filament material for 3D printing must be 95% similar to a vein. The resistance of the artificial lymphatic vessel in the body will depend on it and its acceptance.

The final design of the artificial lymphatic vessel was obtained from the information gathered through the value engineering FAST diagram. The need for a connection between damaged lymphatic vessels to reactivate lymphatic flow was established. For the design test, the pressure and velocity of the lymph in a biological lymphatic vessel were analyzed, evaluating the feasibility of connecting a vessel to the nearest lymph node for its filtering and cleaning. In addition to the connection between lymphatic vessels and nodes, another required function is to prevent the return of the lymph in its unidirectional flow. To achieve this, internal valves were placed in the lymphatic vessel design. Elements of the analysis were behavior, elasticity, pressure, and measurements of valve in the artificial lymphatic vessel.

Three types of materials were analyzed, resulting in the viability of the development of a polymer that could be transformed into a filament for a 3D nano impression and the resistance and effectiveness of its implantation in the body. This, although it represents an annexed work to the one presented, is noted because it will be one of the possible scopes to continue with.

It was possible to design a biological lymphatic vessel, considering the limitation of current manufacturing methods, finally achieving a technique for its nano impression.

Computer fluid simulation allows a successful performance analysis of the proposed artificial lymphatic vessel design; based on the external movements that are required, lymph has correct flow, pressure, and velocity and there is proper vessel function.

The TRL for this project is at level 2 in terms of initial practical applications identified. The potential of the material and the process satisfies the need to restore lymph flow and supports the improvement of lymphedema.

## 6. Patents

The design of the lymphatic vessel is currently in the patenting process at the Mexican Institute of Industrial Property (IMPI), application number IT/B/2021/000537.

## Figures and Tables

**Figure 1 healthcare-10-00068-f001:**
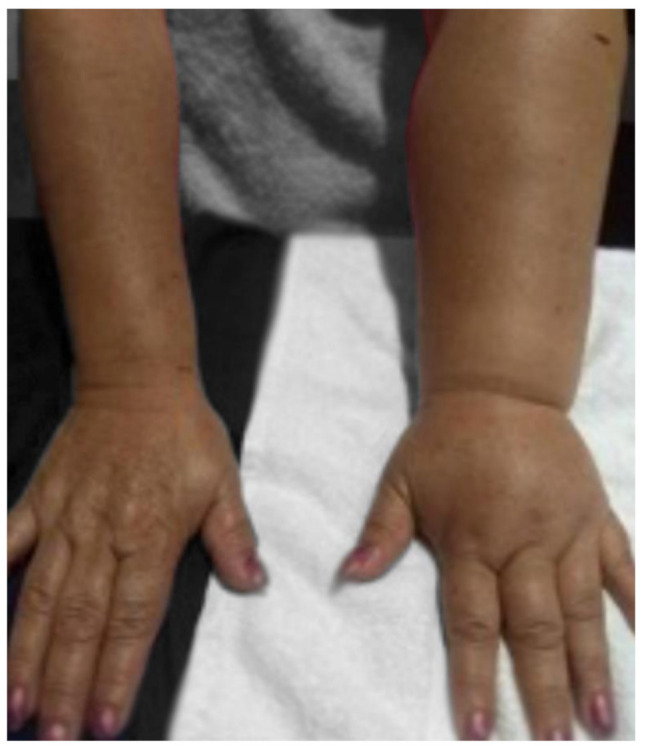
Secondary lymphedema caused by mastectomy with lymphadenectomy.

**Figure 2 healthcare-10-00068-f002:**
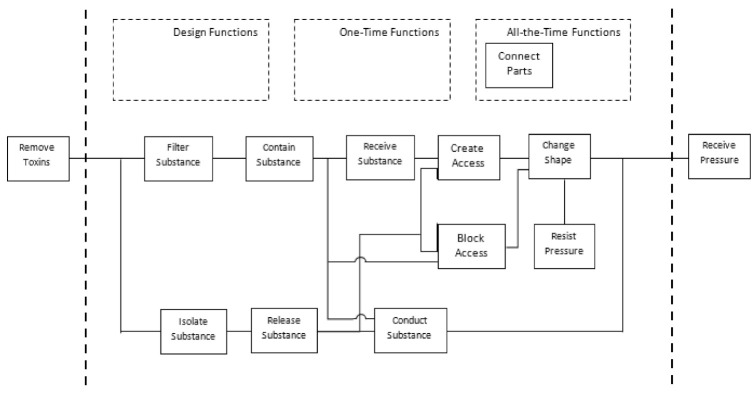
FAST diagram of the device to be designed [26].

**Figure 3 healthcare-10-00068-f003:**
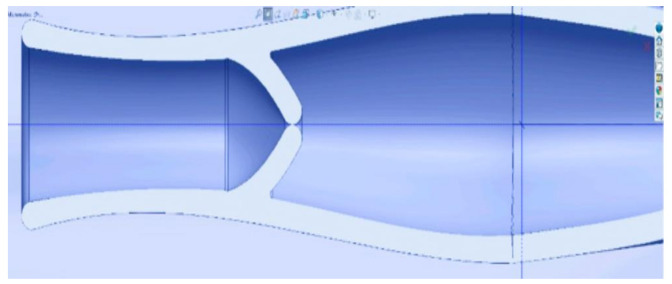
Wireframe model section with a close-up of the lymphatic vessel with a valve similar to the original to prevent the return of lymph (2019.)

**Figure 4 healthcare-10-00068-f004:**
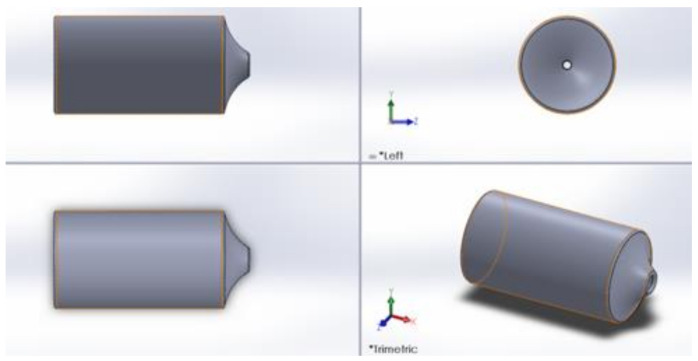
Wireframe model section with a close-up of the lymphatic vessel with a valve similar to the original to prevent the return of lymph (2019).

**Figure 5 healthcare-10-00068-f005:**
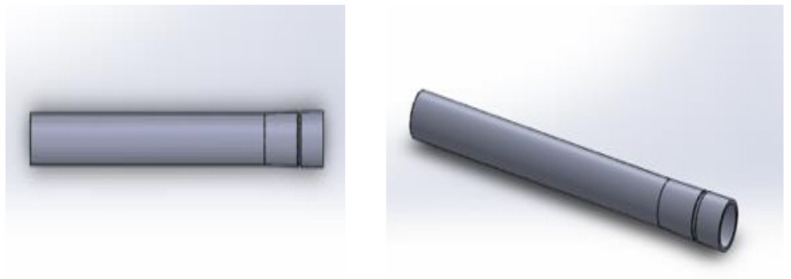
Final version of geometric lymphatic vessel wireframe model with valve at the ends (2020).

**Figure 6 healthcare-10-00068-f006:**
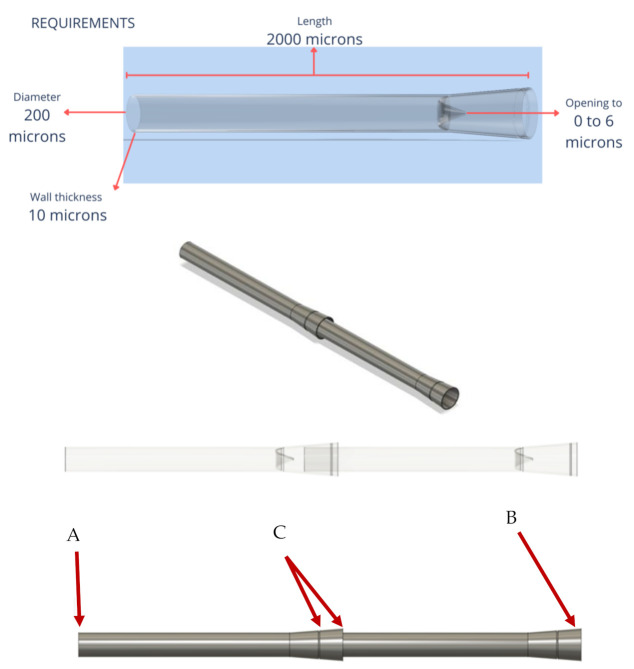
Wireframe model modular lymphatic vessels (2020).

**Figure 7 healthcare-10-00068-f007:**
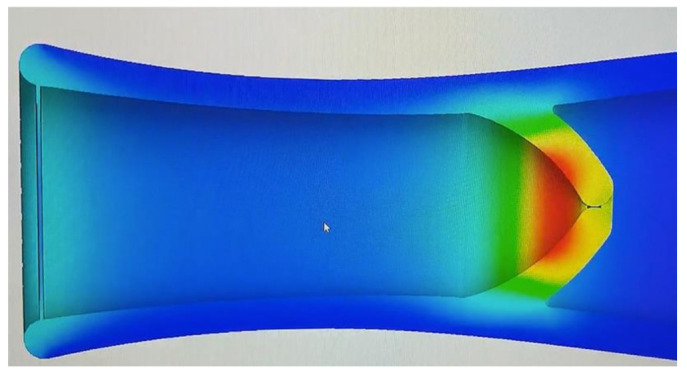
Version 1.0 of the lymphatic vessel: deformation test with open valve (2020).

**Figure 8 healthcare-10-00068-f008:**
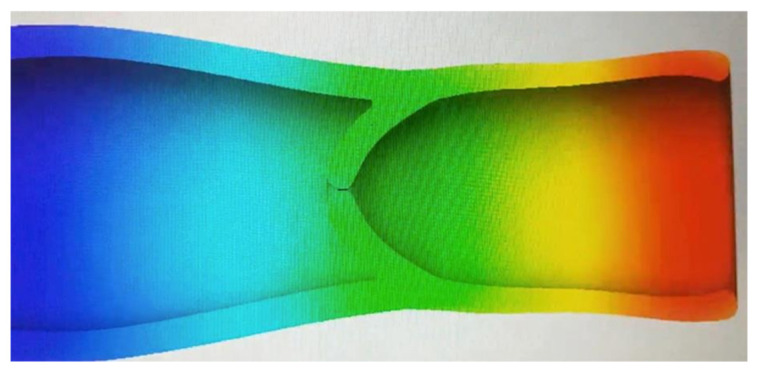
Version 1.0 of the lymphatic vessel: deformation test with closed valve (2020).

**Figure 9 healthcare-10-00068-f009:**
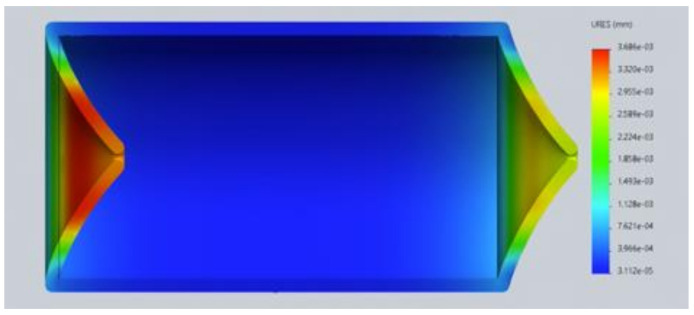
Version 2.0 of the lymphatic vessel: deformation test with open left valve and closed right valve (2020).

**Figure 10 healthcare-10-00068-f010:**
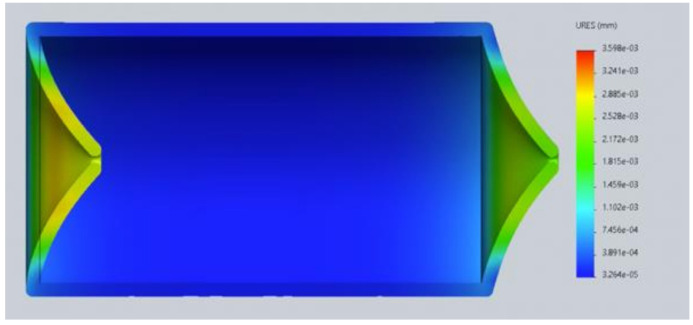
Version 2.0 of the lymphatic vessel: deformation test with both valves closed (2020).

**Figure 11 healthcare-10-00068-f011:**
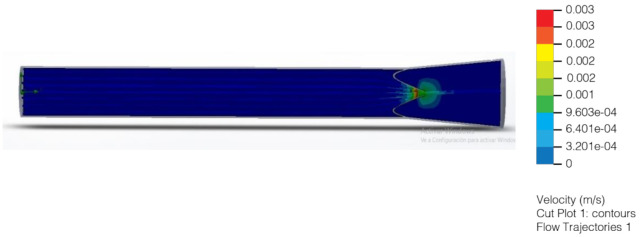
Flow of the liquid in relation to the speed in the inlet and the outlet valves (2020).

**Figure 12 healthcare-10-00068-f012:**
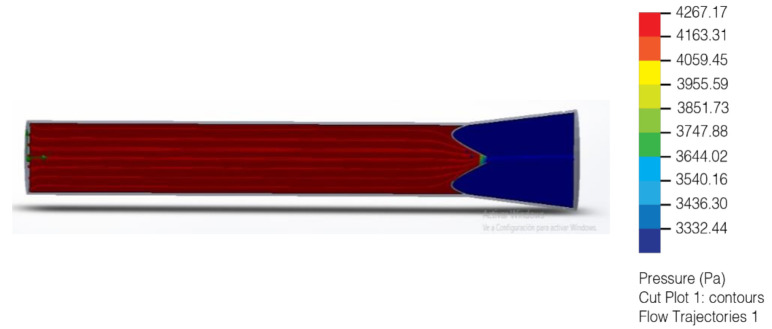
Flow of the liquid in relation to the pressure in the inlet and the outlet valves (2020).

**Figure 13 healthcare-10-00068-f013:**
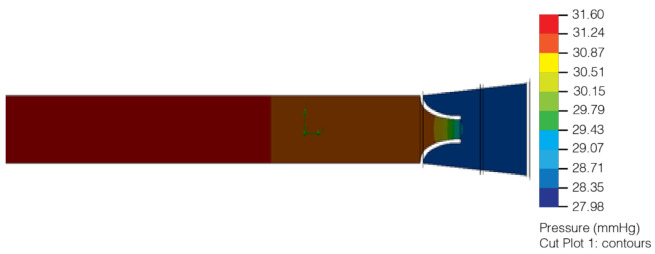
Liquid flow with pressure in open valve; it has a correct one-way direction (2020).

**Figure 14 healthcare-10-00068-f014:**
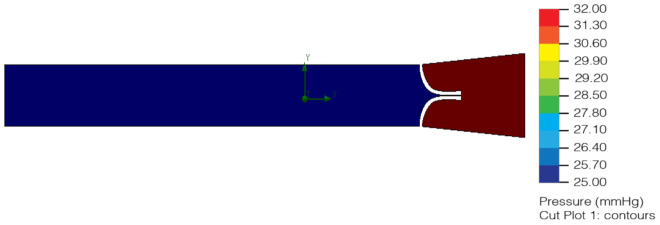
Liquid flow with pressure in closed valve correctly preventing the return of lymph (2020).

**Figure 15 healthcare-10-00068-f015:**
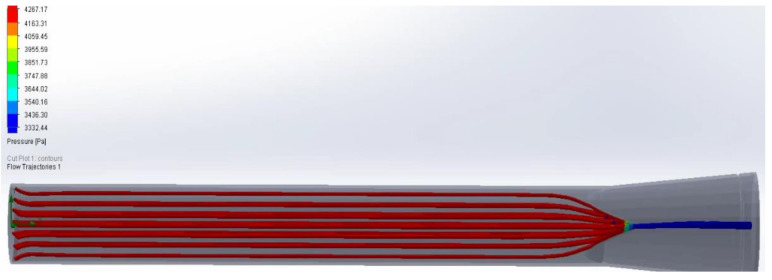
Computer simulation analysis of fluid with pressure of 32 mmHg to 25 mmHg (2020).

**Figure 16 healthcare-10-00068-f016:**
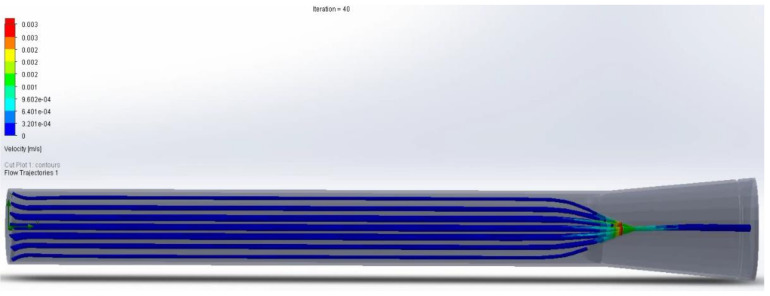
Computer velocity test simulation analysis of fluid with pressure of 32 mmHg to 25 mmHg (2020).

**Figure 17 healthcare-10-00068-f017:**
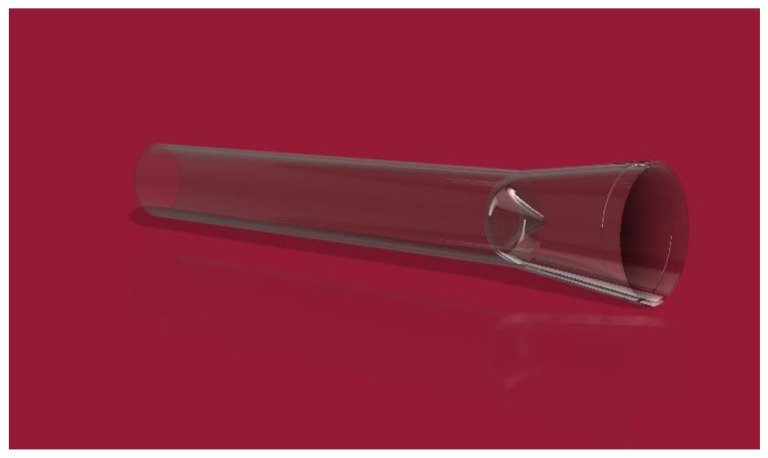
Final lymphatic vessel design (2020).

**Table 1 healthcare-10-00068-t001:** Random list of functions according to the requirements obtained from a lymphatic vessel [26].

Information	Function		
Area	Requirement	Active Verb	Measurable Noun
Geometry	Diameter: 7 mm	Store	Substance
	Volume: 0.4 mL	Store	Substance
		Create	Volume
	Venous Connection	Create	Access
		Connect	Parts
Kinematics	Unidirectional flow	Conduct	Substance
	Valve opening	Create	Access
		Receive	Substance
		Ration	Substance
		Release	Substance
		Block	Access
Forces	Lymph flow depends on external forces, like muscle contractions and joint pumps	Change	Shape
		Resist	Pressure
Material	Medical Grade	Comply	Regulations
		Resist	Bacteria
	Shore A80	Change	Shape
		Receive	Pressure
		Resist	Pressure

## Data Availability

Not applicable.

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
