# Peer review of "Design of an Auxiliary Artificial Lymphatic Vessel in Treatment of Secondary Lymphedema Due to Breast Cancer"

_healthcare, 2021, doi:10.3390/healthcare10010068_

Round 1
Reviewer 1 Report
Thank you for having such a twenty-first century advancement and innovation. I hope it works in human beings. Study design, protocol are okay.
My concerns are: could there be a intolerance of human tissues to that medical device?
Do you already have animal studies to prove the patency of this medical device?
Author Response
Thank you for having such a twenty-first-century advancement and innovation. I hope it works in human beings. Study design, protocol are okay.
Thank you for your kind comments.
1.-My concerns are: could there be an intolerance of human tissues to that medical device?
Selected material is a Thermoplastic elastomer (TPE Mediprene 500M®, Shore 0A from Hexpol® TPE) complies with the mechanical properties for the correct functioning of the lymphatic vessel prototype, as well as with its biocompatibility and medical-grade that guarantees the standards to which it has been submitted.
2.-Do you already have animal studies to prove the patency of this medical device?
No, at this point, we don’t have animal tests, this is the first validation with physicians, showing them some simulations and a 3D print of the prototype to obtain their insights.
Reviewer 2 Report
Primary concerns:
- The wording “prototype” and “resulting in the production of a micrometric prototype” in the title and abstract may be misleading. There is no actual device being made. Consider changing the wording: something like design, modeling, simulation etc.
https://www.merriam-webster.com/dictionary/prototype
a first full-scale and usually functional form of a new type or design of a construction (such as an airplane)
- Missing information on the “treatments the lymphedema”. How would a surgical procedure look like to implement this prototype?
- Lack of description of mechanical properties of the materials being used. Missing description on deformation, stress, and stain in the simulation.
Other concerns/suggestions:
- Image quality of the figures are less than idea. Hard to read the words.
- May need permission to use figures in published work.
- Color bar showing pressure scale is missing.
Author Response
Primary concerns:
- The wording “prototype” and “resulting in the production of a micrometric prototype” in the title and abstract may be misleading. There is no actual device being made. Consider changing the wording: something like design, modeling, simulation etc.
https://www.merriam-webster.com/dictionary/prototype
a first full-scale and usually functional form of a new type or design of a construction (such as an airplane)
Thank you for your comments. You are right, we misleading the title and the abstract, we have changed them for “Design of an auxiliary artificial lymphatic vessel in treatment of secondary lymphedema due to breast cancer.”.
We hope these changes could fit the work we are presenting in a better form.
2.-Missing information on the “treatments the lymphedema”. How would a surgical procedure look like to implement this prototype?
During the design process, we asked multiple clinicians about the node removal surgical procedure and the possibility of using the same traumatic event to position the device. They explain that it may be possible and then we did research on Anastomosis procedure consisting of the connection of two things that are normally diverging. In medicine, an anastomosis typically refers to a connection between blood vessels or between two loops of the intestine.
3.-Lack of description of mechanical properties of the materials being used. Missing description on deformation, stress, and stain in the simulation.
We add the mechanical properties of the materials being used. We also run a simulation to study the deformation of the design with the materials and the stress and strain obtained.
Other concerns/suggestions:
4.-Image quality of the figures are less than idea. Hard to read the words.
We changed the words in the images and improve the quality.
5.-May need permission to use figures in published work.
All the figures are our property.
6.-Color bar showing pressure scale is missing.
You are right, we made the figures again and try to show the pressure scale with a bigger word size.
Reviewer 3 Report
The paper in question is an important report that shows the design method used for the blueprint of the lymphatic vessels and the computer analysis of the fluid simulation and the selection of the proposed materials that led to the production of a micrometric prototype. I was concerned about the following points and recommend revising them.
(1) In the abstract, the author states "This article analyzes changes in breast cancer incidence, mortality, and survival patterns, confirming that, specifically, lymphedema has high health, social and economic impact. Research demonstrates that it fundamentally affects women at an early age" However, the paper did not analyze them. The author just reviewed the information in the paper, so please rephrase the sentences.
(2) The abstract should describe the core content of the study to be reported and should therefore summarise the content of the research development rather than information on lymphoedema.
(3) If you have any concerns about the introduction of the product into animal testing or clinical practice, please describe them.
Author Response
The paper in question is an important report that shows the design method used for the blueprint of the lymphatic vessels and the computer analysis of the fluid simulation and the selection of the proposed materials that led to the production of a micrometric prototype. I was concerned about the following points and recommend revising them.
(1) In the abstract, the author states "This article analyzes changes in breast cancer incidence, mortality, and survival patterns, confirming that, specifically, lymphedema has high health, social and economic impact. Research demonstrates that it fundamentally affects women at an early age" However, the paper did not analyze them. The author just reviewed the information in the paper, so please rephrase the sentences.
Thank you for your comments. We rephrase the sentences in the abstract and also in the introduction. The changes made were: This article reviews breast cancer incidence, mortality, and survival patterns…
(2) The abstract should describe the core content of the study to be reported and should therefore summarise the content of the research development rather than information on lymphoedema.
(3) If you have any concerns about the introduction of the product into animal testing or clinical practice, please describe them.
In order to introduce the product into animal testing or clinical practice, we need to cover some steps first:
- To elevate the TRL of the device. First to make it in the selected materials and run some toxicity tests under several conditions.
- To write a clinical protocol in collaboration with a Laboratory Animal Center and a Medical Specialist and to submit it to an ethics committee, hoping to be accepted.
- To search for grants to pay the test and carry them on once we have the acceptance of the ethics committee.
